# Effects of Five Substances with Different Modes of Action on Cathepsin H, C and L Activities in Zebrafish Embryos

**DOI:** 10.3390/ijerph16203956

**Published:** 2019-10-17

**Authors:** Eberhard Küster, Stefan Kalkhof, Silke Aulhorn, Martin von Bergen, Ulrike Gündel

**Affiliations:** 1Department Bioanalytical Ecotoxicology, UFZ- Helmholtz -Centre for Environmental Research—UFZ, 04318 Leipzig, Germany; 2Department of Molecular Systems Biology, UFZ- Helmholtz -Centre for Environmental Research—UFZ, 04318 Leipzig, Germany; 3Institute of Bioanalysis, University of Applied Sciences Coburg, 96450 Coburg, Germany; 4IZI, Fraunhofer Institute for Cell Therapy and Immunology, Department of Therapy Validation, 04103 Leipzig, Germany; 5Faculty of Life Sciences, Institute of Biochemistry, University of Leipzig, 04103 Leipzig, Germany; 6Department Chemicals and Product Safety, Federal Institute for Risk Assessment (BfR), 10589 Berlin, Germany

**Keywords:** vitellogenin (Vtg), toxicity, risk assessment, 3R animal test replacement method, cysteine protease

## Abstract

Cathepsins have been proposed as biomarkers of chemical exposure in the zebrafish embryo model but it is unclear whether they can also be used to detect sublethal stress. The present study evaluates three cathepsin types as candidate biomarkers in zebrafish embryos. In addition to other functions, cathepsins are also involved in yolk lysosomal processes for the internal nutrition of embryos of oviparous animals until external feeding starts. The baseline enzyme activity of cathepsin types H, C and L during the embryonic development of zebrafish in the first 96 h post fertilisation was studied. Secondly, the effect of leupeptin, a known cathepsin inhibitor, and four embryotoxic xenobiotic compounds with different modes of action (phenanthrene—baseline toxicity; rotenone—an inhibitor of electron transport chain in mitochondria; DNOC (Dinitro-ortho-cresol)—an inhibitor of ATP synthesis; and tebuconazole—a sterol biosynthesis inhibitor) on *in vivo* cathepsin H, C and L total activities have been tested. The positive control leupeptin showed effects on cathepsin L at a 20-fold lower concentration compared to the respective LC_50_ (0.4 mM) of the zebrafish embryo assay (FET). The observed effects on the enzyme activity of the four other xenobiotics were not or just slightly more sensitive (factor of 1.5 to 3), but the differences did not reach statistical significance. Results of this study indicate that the analysed cathepsins are not susceptible to toxins other than the known peptide-like inhibitors. However, specific cathepsin inhibitors might be identified using the zebrafish embryo.

## 1. Introduction

Molecular biomarkers play an important role in chemical toxicity assessments for human and environmental safety [1,2,3]. The effects of contaminants on biomarkers often precede the effects at the morphological level [4,5]. Molecular biomarkers are thought to provide early indications of exposure to specific or unspecific toxic effects within organisms or populations. In addition, bioassays based on molecular biomarkers supposedly have the potential to enable the high-throughput and highly cost-efficient as well as sensitive and specific assessment of chemicals [6,7,8,9,10]. The present study focuses on the investigation of three cathepsins as possible markers of chemical stress in embryos of the zebrafish (*Danio rerio*, Hamilton-Buchanan 1822), an established ecotoxicological and toxicological model organism. Zebrafish embryos are applied in the ‘fish embryo acute toxicity test’ (FET) [11,12,13]. In an earlier study, we analysed the proteome of exposed zebrafish embryos and observed that toxic stress changed the 2D-PAGE protein pattern of vitellogenin-derived yolk proteins [5]. Vitellogenins (Vtg), being phosphoglycolipoproteins, are the major nutrient storage yolk proteins in all oviparous vertebrates. They are enzymatically processed to supply the embryos with essential nutrients at least until external feeding starts [14,15,16]. It was therefore hypothesised [5] that the activity of the enzymes involved in Vtg processing such as cathepsins could be directly or indirectly altered by toxicant exposure or toxicant-derived stress. This would explain the observed change in the Vtg protein pattern in [5] and [17]. 

For many oviparous animals including fish, birds, insects, and amphibians, cathepsin-type proteases are involved in the Vtg and yolk degradation processes [18,19,20,21,22,23,24]. Cathepsins (Cat) are intracellular endoproteases that are mainly known as being responsible for general protein breakdown in lysosomes. In contrast to cellular lysosomal, cathepsins that immediately cleave all proteins into free amino acids, yolk cathepsins are controlled by time- and development-dependent regulation mechanisms [19,21,23] and [25]. The activity or the expression of several cathepsin types has been associated with oocyte maturation and/or embryogenesis—being related to the processing of Vtgs in lower vertebrates, fish and birds. These include aspartic proteases such as CatD as well as several cysteine proteinases such as cathepsins B, C, F, H, K, L, S and Z ([19,20,25] and [26,27,28]). In teleosts, the biochemical information on yolk degradation during embryogenesis is limited to salmonid and pleuronectid species [27] and [29,30,31]. To our knowledge, only a small amount of data on yolk enzymatic processing in zebrafish embryos is available to date: the activities of CatB, D and L have been shown in zebrafish oocytes [32,33] and the expression rate of one cathepsin L gene (Ctsla) and the localisation of CatS- and CatC-mRNA in the yolk syncytium layer of zebrafish embryos were described [28]. 

Different authors have emphasised that an undisturbed yolk nutrition supply and cathepsin activities are vital for normal oogenesis and embryonic development [21,22,26,34]. The term Embryonic Malabsorption Syndrome (EMS) for impairments in the utilisation of endogenous reserves from the yolk was coined by [34]. Since toxic stress has also been shown to change yolk degradation in diverse fish species [14] and also in zebrafish embryos [5], the present study aimed to answer the following three questions. Are cathepsins non-specifically inhibited by toxins (such as polycyclic aromatic hydrocarbons or PAH, insecticides, fungicides) and, if so, can cathepsin enzyme activity changes be used as biomarkers of abnormal yolk degradation in embryonic development after xenobiotic exposure and is this connected to the above mentioned EMS? So far, two studies have concentrated on cathepsins as indicators of fish egg viability [32,33]. The aspartic protease CatD (cathepsin, type D) was altered in eggs inseminated by fish exposed to nonylphenol [32]. However, to our knowledge, cathepsin activities in toxicant-treated zebrafish embryos have not been studied so far.

In the present study, the activities of three cysteine proteases, CatH, CatC and CatL, and their reaction to five xenobiotics were analysed in whole homogenates of exposed zebrafish embryos. These xenobiotics were leupeptin (a known CatL inhibitor as the positive control), phenanthrene, DNOC (dinitro-o-cresol), rotenone and tebuconazole. DNOC and rotenone caused changes in the yolk protein pattern of exposed *Danio rerio* embryos [35]. 

Both DNOC and rotenone affect the respiratory chain but have different molecular toxicity targets [36,37]. Tebuconazole is used as a triazole fungicide with a sterol biosynthesis inhibitor mode of action [38] and phenanthrene was used as a substance with an assumed non-specific mode of action (baseline or narcotic toxicity). The five substances have a more or less known specific mode of action. Still, it is assumed that the induced stress—which is channelling into interconnected “down the river” changes on the metabolism—influences the yolk degradation at some point of the exposure. It is believed that the embryo might increase its energy demand to defend itself against specific anti-oxidative effects or react with a change in metabolism. An increased energy demand might be solved via increased yolk degradation and a change in yolk degradation enzyme activities such as those of cathepsins. 

By measuring cathepsin activities in whole embryo homogenates, extracellular yolk cathepsin enzyme activities might be confounded by concurrent intracellular cathepsin enzyme activities.

The differentiation between extra- and intracellular cathepsin activities in fish embryos would be crucial for toxicity studies on both cathepsin groups. However, such time-consuming separation is not compatible with the present objective of developing a simple, time-saving and validated method of analysis. 

Hence, while being focussed on yolk cathepsin activities, the outcome of the study integrates information on the possible xenobiotic impact on the activities of other (intracellular) cathepsins and all measured non-specific enzymatic reactions could contribute to yolk processing such as vitellogenin degradation. Therefore, we think that non-specific enzymatic reactions should be similarly considered when cathepsin activities are consulted to indicate abnormal yolk degradation processes unless specific reactions can be proven.

## 2. Materials and Methods 

### 2.1. Fish Cultivation and Fish Egg Collection

Wild Indian Karyotype (WIK) [39] zebrafish embryos (*Danio rerio*) were obtained from the Tübingen Zebrafish Stock Centre (recently transferred to the European Zebrafish Resource Center, http://www.ezrc.kit.edu/) and cultivated as described in [40], resembling standard protocols published in the “zebrafish book” [41], in [42] and in the OECD Guideline [13]. Within 1 h of spawning, fish eggs were collected, cleaned of faeces and debris and controlled for fertilisation. Fertilised eggs in the four- to eight-cell stage were incubated in aerated standard dilution water, i.e., ISO-water [43], as described in [44], with a density of one egg per 2 mL at static conditions (27 ± 1 °C, 12/12 h light/dark, and at 30 µmol photons m^−2^ s^−1^). Developmental stages were distinguished according to [45] using an inverse microscope (Olympus IX70-S8F, Hamburg, Germany) at 50x magnification.

### 2.2. Fish Embryo Toxicity Test (FET)

The fish embryo acute toxicity test for *Danio rerio* embryos (FET) [11] was carried out with five substances (leupeptin (Chemical Abstracts Service Registration Number or CAS RN 24365-47-7, Serva, Heidelberg, Germany), DNOC (4,6-dinitro-o-cresol, CAS RN 534-52-1, Riedel-deHaen, Seelze, Germany), rotenone (CAS RN 83-79-4, Riedel-deHaen, Seelze, Germany), tebuconazole (CAS RN 107534-96-3, Riedel-deHaen, Seelze, Germany) and phenanthrene (CAS RN 85-01-8, Sigma-Aldrich, Seelze, Germany)). The test was performed as recommended in OECD Guideline 236 for the Testing of Chemicals [13]. Assays for leupeptin and DNOC were performed in polystyrene 24-well microtiter plates (Greiner GmbH, Frickenhausen, Germany) using 10 embryos for each concentration. Due to their lipophilicity (log Kow above 3) and the consequential possible substance loss [46,47], the FET for rotenone and tebuconazole was carried out in glass Petri dishes covered with glass lids (also 10 embryos per concentration). No solvents were used in any of the FET tests. 

The test concentrations were selected due to their max. water solubility (see Table 1) and either from literature data or from their modelled baseline toxicity (Table 1). Thus, test concentration ranges were selected to be between 0.002 and 0.2 µmol/L for rotenone, 0.07 and 0.6 µmol/L for DNOC, 0.06 and 12 µmol/L for phenanthrene, 9 and 80 µmol/L for tebuconazole and 10 to 1000 µmol/L for leupeptin.

After 48 h of exposure, embryos were screened for the four lethal effects (coagulation, missing heartbeat, missing tail/yolk detachment and missing somites, OECD 2013). Concentration–effect relationships were modelled based on the sum of the abovementioned lethal effects and using the three parameter sigmoidal Hill function f = a*x^b/(c^b+x^b) (with a = 100, c = LC_50_ and b = slope). Data fitting was carried out with the software SigmaPlot™ Version 12 (Systat Software Cooperation). 

### 2.3. Danio Rerio Embryo Collection for Enzymatic Measurements during Development

For the enzyme analysis, 30 embryos were pooled for each stage analysed and this was repeated 2 times. Thus, each of the seven embryonic stages (see below) was analysed for enzyme activities with 90 embryos altogether (630 embryos for all the enzyme tests). The following stages were selected: 0.5 hpf (zygote period), 2.5 hpf (blastula period), 9 hpf (gastrula period), 26 hpf (pharyngula period), 49 hpf (long-pectoral stage in hatching period), 73 hpf (protruding-mouth state in hatching period) and 97 hpf (early larval state), and each group of 30 embryos per stage was collected in 2 mL Eppendorf ^®^ reaction tubes. The embryos were then washed two times with 1 mL of *aqua bidest* and immediately frozen in liquid nitrogen. 

### 2.4. Exposure Conditions to Study In Vivo Effects on Cathepsin Activities

The cathepsin enzymes reached highest activities in approximately 48 to 72 h. As we wanted to be able to compare toxicity data to our own results and to other publications, we used 48 h as the maximal exposure duration and did not increase it any further. 

Zebrafish embryos (n = 70–90), up to 2 h post fertilisation (64-cell stage), were incubated collectively for 48 h in the test solution under static conditions (leupeptin, DNOC, rotenone, tebuconazole or phenanthrene) and in ISO-water, e.g., negative control in glass Petri dishes with a density of one embryo/2 mL. After 48 h of exposure, all embryos were screened for morphological effects and 60 intact embryos, i.e., not showing visible detrimental effects, were pooled in one reaction tube. Each of these “60-embryo-pools” is referred to as one replicate. Embryos were washed at least 6 times with 1 mL of bi-distilled water before freezing in liquid nitrogen and stored at −80 °C. For the analysis of the concentration dependence on the cathepsin activities, one replicate per concentration was exposed in the experiment. For the low-dose experiments, nine replicates that were exposed under the same conditions but used on different days or by different experimenters were used for the cathepsin studies. 

### 2.5. Protein Extraction from Danio rerio Embryos

Homogenisation of the frozen embryos was carried out on ice in 10 µL of lysis buffer (0.1 M Na acetate, 1 mM EDTA, 0.1% triton x-100, pH 4) per embryo (e.g., 300 µL for 30 embryos) using a metal-bladed homogeniser (UltraTurrax, IKA Labortechnik, Staufen, Germany) for 30 s at a speed of 300 rpm. The homogenates were centrifuged at 4 °C for 30 min at 9000× *g* (Centrifuge 2K15, Sigma, Osterode, Germany) and the supernatant was stored on ice for a maximum of 30 min before analysis of the sample. Protein concentration was measured according to [48], using the BIO-RAD DC protein assay (BIO-RAD, Munich, Germany). Furthermore, 1D SDS-PAGE (sodium dodecyl sulfate–polyacrylamide gel electrophoresis) analyses were carried out on 12.5% polyacrylamide gels using the method of [49] and an ESI–MS/MS-based identification of proteins of interest was performed as described in the Appendix A with an LTQ-Orbitrap mass spectrometer.

### 2.6. Cathepsin Assays (CatH, CatC, CatL)

The cathepsin assays were carried out in black 96-well microtiter plates (Art.#701334, Brand Life Science Products, Wertheim, Germany). All assays were based on the conversion of cathepsin-specific substrates to the fluorogenic product 7-amino-4-methylcoumarin (AMC, Calbiochem) and thus enabled the same enzymatic conditions for all analysed cathepsin activities. Fluorescence was measured in a kinetic study every 2nd min for 60 min after starting the assay reaction using a microplate spectrofluorometer (Spectra Max Gemini, Molecular Devices, Sunnyvale, USA) at 360 nm excitation and 460 nm emission, at 37 °C. All assays were carried out in 200 µL of assay solution according to the InnoZyme^TM^ Cathepsin L activity kit (Calbiochem, Gibbstown, USA) in four steps. At first, 50 µL of assay buffer (0.4 M acetic acid, 4 mM Na EDTA, 0.8% TCEB, pH5.5) and, secondly, 50 µL of either embryonic protein sample, lysis buffer (see above) as chemical blank, AMC standard or pure cathepsins were added into the wells. The plates were sealed with a self-adhesive plastic film and gently shaken for 15 min at room temperature (RT). In a third step, 50 µL of the diluent 0.1% Brij^®^ 35 detergent (Calbiochem) (e.g., blanks and total substrate conversion) were added. The plate was gently shaken again for 15 min at RT. Fourthly, the reaction was started by adding 50 µL of 0.1 mM substrate solution. Each sample was measured twice, with and without addition of a specific CatL/H inhibitor (third step above). Specific activity was then obtained according to Equation (1).
(1)$$S_{\mathrm{Total}}\;=\;S_{\mathrm{Specific}}\;+\;S_{\mathrm{Unspecific}}$$
where S_Specific_ is the specific substrate conversion, S_Total_ is the total substrate conversion = substrate conversion without inhibitor, and S_Unspecific_ is the non-specific substrate conversion = substrate conversion after addition of inhibitor.

Measurement of cathepsin L activity. CatL activity was assayed using the Z-Phe-Arg-AMC substrate (CAS RN 65147-22-0, Calbiochem). To avoid interference with CatB, the specific CatB inhibitor CA-074 (1 µM final concentration, CAS RN 134448-10-5, Calbiochem) was added to the activation buffer. Z-Phe-Tyr-(tBu)-diazomethylketone (CAS RN 114014-15-2, Alexis Biochemicals, Lörrach, Germany) was used as a specific CatL inhibitor in a final concentration of 2.5 µM.

Measurement of cathepsin H activity. CatH activity was measured by using the fluorogenic compound H-Arg-AMC (CAS RN 70274-89-4, Bachem, Weil, Germany). As a specific CatH inhibitor, H-Leu-chloromethylketone (CAS RN 54518-92-2, Bachem, Weil, Germany) was used in a final concentration of 25 µM.

Measurement of cathepsin C activity. For CatC activity determination, H-Gly-Arg-AMC (CAS RN 65147-19-5, Bachem) was used as a specific substrate. No additional specific or non-specific inhibitors could be used for differentiation as none were commercially available. 

Activities were calculated as units (U) (conversion of 1 µmol/L substrate per minute) and activity data were normalised to protein content in the sample (U/mg total protein). For comparison of cathepsin activities at different developmental stages, data were not normalised to total protein amount in the samples because the protein content in the eggs decreases during embryonic development [50], which could lead to a misinterpretation of the results mainly at lower enzyme activities such as for CatC or CatH. 

The three-parameter sigmoidal Hill function f = a*x^b/(c^b+x^b) (with a= 100, c = EC_50_ and b = slope) was used for modelling concentration–effect relationships. Data fitting was carried out with the software SigmaPlot™ Version 12 (Systat Software Cooperation). 

### 2.7. Ethical Consent

Fish were cultured and used according to German and European animal protection standards and fish culture was approved by the Government of Saxony (Landesdirektion Leipzig, Aktenzeichen 75 ± 9185.64). According to the “EU Directive 2010/63/EU on the protection of animals used for scientific purposes”, early life stages of zebrafish are not protected as animals until the stage of being capable of independent feeding (5 days post fertilisation). All embryos were killed right after the end of the 96 h tests by the accepted methods of the German regulatory bodies.

## 3. Results

### 3.1. Cathepsin Activities in Embryonic Stages of the Zebrafish

Preceding the toxicity-related enzyme assays, baseline cathepsin activities during early zebrafish development at seven different embryonic stages shortly after egg fertilisation (0.5 hpf) until early larval stage (97 hpf) were studied (Figure 1). For this, the CatH, CatC and CatL total substrate conversion rates in fish embryo samples under control conditions were characterised. 

For CatH and CatL substrates, next to specific enzymatic conversion (CatH and L) also non-specific conversion processes that could not be clearly differentiated from each other were observed in the embryo samples (data not shown). Hence, the sums of both total conversion rates (Figure 1) are shown for these substrates only. For the CatC substrate, no non-specific conversion was observed. This result follows data by [51], who have shown that cathepsin C substrate is very specifically converted by dipeptidylaminopeptidase I (CatC). 

Overall, for all measured substrate conversion processes, only minor enzymatic activities were detectable in the early embryonic stages from 0.5 to 9 hpf (90% epiboly). A strong (at least 10-fold) increase in almost all enzyme activities occurred in the segmentation period between 9 and 26 hpf (Figure 1). CatC substrate conversion subsequently remained constant, CatL substrate conversion showed the highest peak shortly after hatching at 74 hpf and strongly decreased until 97 hpf (to an activity resembling 10% of the activity at 74 hpf). In contrast, the total CatH substrate conversion more or less constantly increased during the embryonic development. 

The total CatL and CatC substrate conversion rates are much higher compared to CatH activity (at least 1 to 3 orders of magnitude) at most analysed developmental stages. At all developmental stages, CatL exerted the predominant enzymatic activity. 

### 3.2. Toxicant Effects on Zebrafish Embryo Mortality and on Cathepsin Activities after 48 h of Exposure

#### 3.2.1. Leupeptin Exposure as a Positive Control for Direct in vivo Cathepsin Inhibition

As a first step to characterise the potential of cathepsins as possible biomarkers of toxic stress, embryos were exposed to the CatL inhibitor leupeptin. This substance was chosen as a reference to (i) analyse whether direct cathepsin inhibition can be detected with the established analysis procedure, (ii) characterise the lethal effects of specific cathepsin inhibition and (iii) analyse whether effects can be related to yolk nutrition, e.g., decomposition or digestion. 

Fertilised eggs were exposed to the toxicant for 48 h. Leupeptin caused multiple effects, such as impaired cardiovascular function, pericardial oedema, retardation of development and coagulation. Figure 2A shows the modelled concentration–effect relationship for lethal endpoints according to FET (LC_50, FET_ approximately 0.35 mM). To test the sensitivity of CatL activity to the specific inhibitor leupeptin, five low-effect concentrations between 0.001 and 0.2 mM (23–460 times below the LC_50_) were chosen. For all concentrations, a decrease in CatL activity compared to controls could be detected. The resulting concentration–effect relationship is displayed in Figure 2A. The estimated EC_50, CatL_ value of 0.019 mM is approximately 24 times lower than the LC_50_ determined from the FET assay. Comparing the more sensitive low-effect concentrations LC_20_ (LC_20 FET_ = 0.20 mM, EC_20 CatL_ = 0.0067 mM), the difference increases slightly to approximately 30 times. High inhibiting effects on CatL could already be detected when only a few embryos (1–2) showed effects with the FET. For example, approximately 95% inhibition of CatL was observed at 0.1 mM leupeptin causing phenotypic effects in approximately 10% of the embryos (LC_10, FET_). Moreover, for 0.1 mM (LC_10, FET_) and 0.2 mM (LC_20, FET_) leupeptin, strong effects on the 1D proteome pattern of exposed embryos could be detected by 1D SDS-PAGE (Figure 2B). In comparison to controls, two protein bands (indicated by arrows) at approximately 118 and 66 kDa appeared with a significant higher abundance in the proteome of leupeptin-exposed embryos. These two proteins were identified by ESI–MS/MS analysis. Peptides of both proteins significantly matched to *Danio rerio* vitellogenin 1 (Vtg) sequences. Results are summarised in the Appendix A. The results strongly point to interruptions of vitellogenin usage/degradation of leupeptin-treated embryos.

#### 3.2.2. Effects of Phenanthrene, Rotenone, DNOC and Tebuconazole

Following the leupeptin experiments, the effects of the four xenobiotic chemicals, phenanthrene, rotenone, DNOC and tebuconazole, on embryonic zebrafish cathepsin H, C and L activities were investigated. Before taking the cathepsin activity measurements, the lethal effects of the four substances on the fish embryos were studied after 48 h of exposure. The resulting concentration–effect relationships are depicted in Figure 3.

The range of concentrations tested spanned five orders of magnitude. Rotenone was the most and leupeptin was the least toxic substance analysed. The order from highest to lowest toxicity was rotenone > phenanthrene = DNOC > tebuconazole > leupeptin, with LC50 values ranging from 0.04 to 350 µM (for data for all substances, see Table 1, with LC50 = c = x50). 

Based on these lethal effect results, low exposure concentrations for the cathepsin enzyme analyses were selected. Concentration series with at least five data points for each chemical were tested at concentrations resembling the LC50, LC20, LC10, LC05 and LC01. For all selected low effect concentrations, the CatH, CatC and CatL substrate conversion rates were measured. All observed concentration-dependent effects on cathepsin substrate conversion rates and resulting data fitting are depicted in Figure 4A–D. Concentration-dependent effects on cathepsin substrate conversion rates were found for all tested chemicals except DNOC. As embryos that showed phenotypes that were different from the control were excluded from the samples for the enzymatic measurements, alterations in cathepsin activities could thus be detected in phenotypically intact organisms. Rotenone caused a concentration-dependent inhibition of total CatH substrate conversion (Figure 4B). DNOC led to a slightly higher (but not significant) efficacy in the inhibition of CatH substrate conversion processes compared to rotenone (Figure 4A). The fungicide tebuconazole also showed a concentration-dependent increase in inhibition with CatH and to some extent this was also observed with CatC (Figure 4C). Phenanthrene only affected the activity of CatH to a small extent (Figure 4D), though this was not statistically significant. The other two enzymes, CatC and CatL, did not show any effects under these test conditions (with the exception of rotenone). Although some of the enzymes were concentration-dependently inhibited, overall, all enzyme activity inhibitions were not significantly different compared to lethal effects. The cathepsin activities analysed here are not more sensitive than the observed lethal effect in the fish embryos. Thus, at least these enzymes do not seem to be useful biomarkers of sublethal stress evoked by the four xenobiotics. This is in contrast to the specific CatL inhibitor leupeptin which showed effects on CatL activity at concentrations approximately 20 to 30 times below lethality. What we can conclude from these data is that cathepsin H (and to some extent CatL) seemed to be a “co-marker” for lethality (see discussion below). All four xenobiotics did affect cathepsin activity by 20% to 80% at the same concentrations at which lethal effects occurred. These observed cathepsin inhibitions might be a side effect of dying embryos. In other words, the cathepsin CatL was quite sensitive to the specific inhibitor leupeptin only, while the other xenobiotics did not show any significant effects on CatL or the other cathepsins. 

## 4. Discussion

Different authors have emphasised that regular yolk nutrition—mediated by proteases of the cathepsin type—is vital for normal fish embryonic development [21,22,26,34]. Also, in a previous study, we related toxic stress to impaired yolk consumption processes in zebrafish embryos [5]. The present study therefore aimed to investigate the potential of certain cathepsins to serve as markers [53] to indicate disturbed yolk consumption or nutrition and thus explain impairments in embryonic development caused by xenobiotic exposure. Others have reviewed and proposed various attributes that candidate biomarkers must have [10]. Hence, with reference to these requirements, the data from this study will be discussed concerning biological and methodological validation as well as toxicological aspects. 

### 4.1. Methodological and Biological Validation of Measured Cathepsin Activities

Cathepsins are known to exert major enzymatic activities in cellular lysosomes for normal cellular protein turnover [54,55]. As we have not investigated the spatial distribution of cathepsin activities in the developing embryos, extracellular yolk enzymatic activities and intracellular lysosomal functions of cathepsins cannot be discriminated. The spatial separation of extra- and intracellular cathepsin activities in fish embryos would require the separation of the embryos from yolk vesicles, which would still not remove the vitellogenin completely [17] and [56], or require the use of immuno-cytolocalisation techniques [23]. Moreover, evidence points to the existence of cathepsin enzymes in the yolk. This includes the appearance of cathepsin activities in the eggs shortly after fertilisation. Many studies have shown that cathepsin activities play an important role in fish oocytes before fertilisation when no embryonic cells exist [21,23,57]. Also, mRNA of CatL and CatC was detected in the yolk syncytium layer of zebrafish embryos [25]. Results of the present study have shown a direct correlation between the specific cathepsin L inhibitor leupeptin and impairments in vitellogenin degradation (i.e., yolk mobilisation/consumption). Thus, cathepsin activities measured in whole embryonic homogenates appear to be associated with the yolk rather than with cellular lysosomes as was also assumed by Carnevali and colleagues [26]. An influence of intracellular lysosomal cathepsins on results of the present study cannot be excluded since Kaivarainen et al. [55] have demonstrated the influence of toxic factors for internal proteinases activities. Hence, the cathepsin assay used here considers the entity of the cathepsin H, C and L endoproteases and integrates different enzymatic functions of cathepsins that might all be affected by toxic stress in fish embryos. 

To distinguish between natural variability and contaminant-induced stress, baseline data for the activity of candidate biomarkers under different parameters (temperature, organism age etc.) are needed [10]. So, cathepsin activities at different stages of embryonic development were studied. For the teleost *Danio rerio*, the enzymatic role of cathepsins in yolk processing has not been characterised so far. To our knowledge, this is the first description of the three cathepsins CatH, CatC and CatL during the first 96 h of the embryogenesis of zebrafish. Our findings are supported by the results of Zhang et al. [33], demonstrating some CatL activity in early oocytes of zebrafish, and Tingaud-Sequeira and Cerda [28], who observed mRNA of CatL and C in the yolk syncytial layer of zebrafish embryos. 

All measured cathepsin activities were very low during the first 10 h post fertilisation, which is in accordance with previous results of [5] assuming lower enzymatic activities in the early blastula and gastrula stages. The strong increase in CatH, C and L substrate conversion rates occurring in the segmentation period until 26 hpf is consistent with dramatic changes in the yolk protein pattern of zebrafish embryos during that period of development [5]. A reason for this might be the beginning of embryonic organ and somite formation, leading to the altered energy and nutrient metabolism needs of the embryo [26,45]. In the following developmental stages, all measured cathepsin activities showed different time-dependent progressions. Our findings point to specific developmental enzymatic processes in yolk degradation that were also described by others [19,22,23,25,58,59]. The observation of the activity of more than one cathepsin type and their different activity rates support the hypothesis that yolk protein degradation is realised by various enzymes that might therefore have indicative potential for sublethal stress.

Cathepsins belong to a conserved and homologous enzyme family [25] and therefore non-specific substrate conversion processes cannot be ruled out. Non-specific substrate conversion was detected for CatH and CatL substrates. This could be either due to homologous cathepsins in the samples or the presence of other proteases such as aminopeptidase B which is known to unspecifically cleave CatH substrate [60]. CatC substrate (H-Gly-Arg-AMC) conversion is highly specific for dipeptidyl aminopeptidase I (CatC) [51] and hence a non-specific reaction was neither expected nor detected for CatC. 

Therefore, we think that non-specific enzymatic reactions should be similarly considered when cathepsin activities are consulted to indicate abnormal yolk degradation processes unless specific reactions can be proven.

### 4.2. Toxicity-Related Changes in Cathepsin Activities

Toxicity-related changes in cathepsin activities were studied using the five substances, leupeptin, rotenone, DNOC, phenanthrene and tebuconazole. Leupeptin is a known CatL inhibitor [61] and was hence chosen as a reference toxicant to validate the applied system and to test for effects in fish embryos resulting from direct cathepsin inhibition. Leupeptin had the highest potency and efficacy in inhibiting enzyme activity in relation to lethal effects compared to the other four tested substances. The inhibition of CatL was already detectable at a 20- to 25-fold lower exposure level compared to visible phenotypic effects and a clear relation to changed yolk protein pattern could be shown. The observed low lethality after inhibition of CatL, which was found to exert predominant activity, is astonishing. This result could mean, on the one hand, that cathepsin activity measurements may very specifically only detect the inhibition of cathepsin enzymes. On the other hand, it could mean that substances which do have other modes of actions such as rotenone or DNOC do not directly affect cathepsins and thus would be clearly differentiable to cathepsin inhibition as the mode of action. Still, the data can be interpreted as a direct relationship between cathepsin inhibition, yolk protein degradation and organism health. Results from leupeptin-treated rats showed the interruption in the nutrition of developing conceptus from the visceral yolk sac, which would support our observations [62].

Rotenone and DNOC, two pesticides, exert specific modes of action such as inhibition of the complex I in the electron transport chain and as a decoupler via the inhibition of oxidative phosphorylation, respectively [36,37]. They were chosen for this study because direct links between energy metabolism and yolk consumption including cathepsin activities were assumed. Phenanthrene was chosen as a narcotic reference substance, whereas tebuconazole was chosen as a substance with a known specific effect (sterol biosynthesis inhibitor) but suspected low effect on yolk degradation via inhibition of cathepsins.

Effects on cathepsin activities were detectable with all four substances. The effects on cathepsin activities increased with rising toxicant concentrations, demonstrating concentration dependence. However, the differences between enzyme activities and visible lethal effects were not as great as expected. The study thus shows that at least the cathepsins analysed here (H, C and L) did not have any or only little marker potential for signalling toxic exposure or effect by non-cathepsin inhibitors (at least when considering the tested substances and their mode of action). Hence, it is concluded that cathepsins H, C and L are unlikely to indicate toxic exposure/effects on the molecular level sensitively at lower concentrations, at least if substances do not specifically inhibit these cathepsins. So far, no firm conclusions on chemical specificity can be drawn. However, differences in the pattern of cathepsin interference caused by the different toxicants were noticeable with respect to the type of cathepsins being affected, the number of affected cathepsin types, the induction or inhibition of enzymatic activities, and also regarding differences in potency and efficacy. The biochemical interpretation of these results cannot be taken any further, because the specific roles of cathepsins during embryonic development are not completely understood, but they support the assumption that different toxicants might affect cathepsin activities. Future research could focus on systematising the specific types of exposure and effects that the different cathepsins may reliably indicate. 

To our knowledge, xenobiotic effects on embryonic cathepsins in exposed fish embryos have not been investigated, but our results are in line with the few studies that have tested the indicative potential of cathepsins in different fish developmental stages so far. Others, such as [32] and [33], detected impaired embryonic CatD activities in eggs from nonylphenol-exposed fish and investigated embryonic CatB, L and D activities as markers for successful cryoprotection in zebrafish embryos [33]. Moreover, embryonic CatL activity was associated with follicular atresia in rainbow trout [63]. Some studies have shown the potential of cathepsins as biomarkers for xenobiotics in adult fish [53,64]. They have introduced CatD as a biomarker for endocrine disruption, and Kaivarainen et al. [55] demonstrated the xenobiotic effects of ore processing wastewater on intracellular cathepsin activities in adult fish.

## 5. Conclusions

Many authors have suggested the indicative potential of embryonic cathepsins in the monitoring of disturbances in embryonic development. The present study tested this hypothesis by analysing three embryonic cathepsins as biomarkers for xenobiotic stress in zebrafish embryos. The experiments with four xenobiotics and one specific and known cathepsin inhibitor show that the measured cathepsins do not seem to exhibit the sensitivity attribute requested for candidate biomarkers [10]. In addition to this baseline data, concentration dependence, easy measurement methods and a clear relation to developmental disorders and organismic health was observed. Thus, the results of the present study do not support the use of cathepsins H, C and L as general biomarkers for monitoring embryonic development and xenobiotic exposure in oviparous embryos, but they do support their use as biomarkers for identifying cathepsin inhibition as the specific mode of action. Furthermore, future research could focus on investigating whether other embryonic cathepsins are suitable biomarkers. Future research might concentrate on the investigation of spatial distribution and the functions of different cathepsin types, as well as systematise and characterise their sensitivity and specificity for different toxicant groups and study the transferability of findings to other oviparous organisms.

## Figures and Tables

**Figure 1 ijerph-16-03956-f001:**
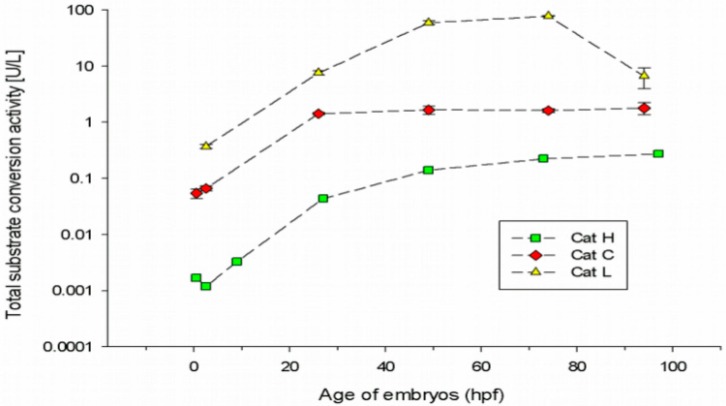
CatH (■), CatC (♦) and CatL (▲) total substrate conversion activity at five to seven different time points (0.5, 2.5, 9, 26, 49, 73 and 97 hpf) during the early development of zebrafish embryos. For each data point, ± SD of three replicates (30 fish embryos per replicate) are shown.

**Figure 2 ijerph-16-03956-f002:**
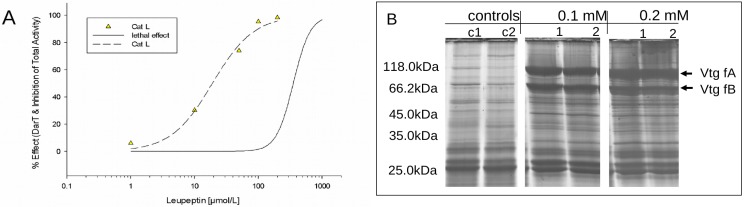
Effect characterisation after 48 h of exposure to different concentrations of leupeptin to zebrafish embryos. (**A**) Concentration–effect relationship for the assessment of lethal effects (solid line) and effects on CatL activity are shown (model and curve parameters please see Table 1). (**B**) SDS-PAGE from protein samples of total embryo homogenates of untreated (controls) and leupeptin- treated (100 µM and 200 µM) embryos. Two replicates are shown, each deriving from 30 intact-looking embryos. The arrows indicate two proteins that very obviously changed compared to controls. They were identified as vitellogenin 1 (Vtg) fragments (f) by ESI–MS/MS-based identification using an LTQ-Orbitrap mass spectrometer (please see Appendix A for more information).

**Figure 3 ijerph-16-03956-f003:**
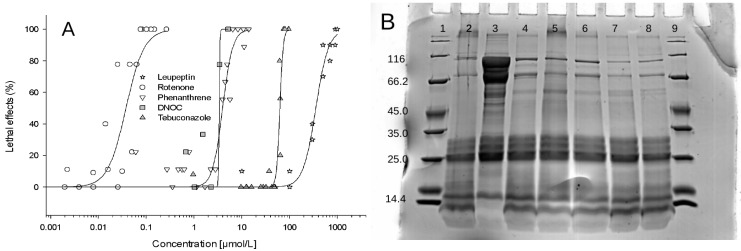
Concentration–effect relationships of the lethal effects of the five test substances, rotenone, DNOC, phenanthrene, tebuconazole and leupeptin, on zebrafish embryos after 48 h of exposure (**A**). Concentration–effect relationships were modelled using the three-parameter sigmoidal logistic function (Hill) f = a*x^b/(c^b+x^b) (for parameters, see Table 1). Each data point represents the mean of 10 embryos of the tested concentration. (**B**) SDS-PAGE from protein samples of total embryo homogenates of untreated (controls) and treated embryos. Treatment included the exposure to the respective LC_10_ concentrations of all five test substances for a duration of 48 h. Lanes 1 and 9: protein marker, 2: control, 3: leupeptin, 4: roteone, 5: DNOC, 6: phenanthrene, 7: tebuconazole, and 8: DMSO control.

**Figure 4 ijerph-16-03956-f004:**
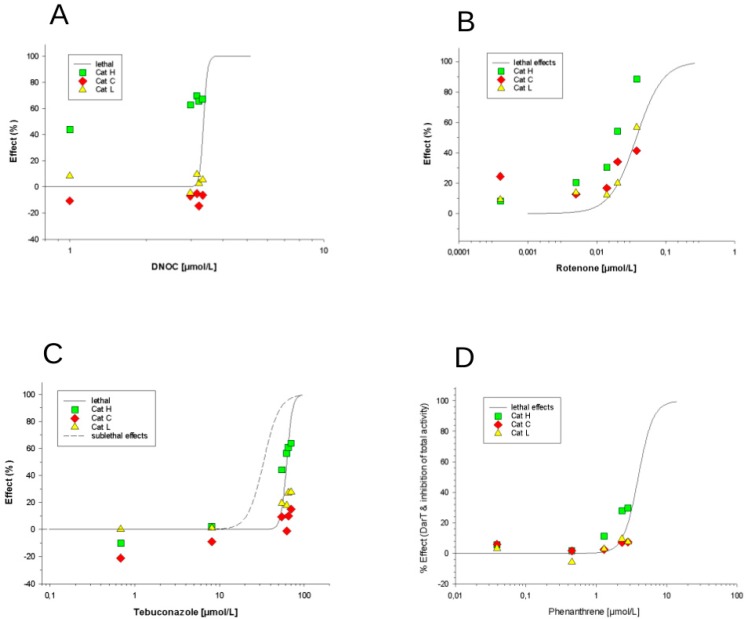
Effects of (**A**) DNOC, (**B**) rotenone, (**C**) tebuconazole and (**D**) phenanthrene on cathepsin activities in zebrafish embryos. CatH, C and L substrate conversion activities were measured at different toxicant concentrations and only concentration-dependent effects are presented. All data points originate from homogenates of 60 intact embryos, not showing any phenotypic lesions. Data are shown in relation to observed phenotypic effects obtained by FET (solid lines). For parameters, see Table 1.

**Table 1 ijerph-16-03956-t001:** Chemical identities, physicochemical properties and model parameters from concentration–effect relationships established for lethal (FET) and sublethal effects after the exposure of zebrafish embryos to the five test substances.

	Leupeptin	Rotenone	DNOC	Phenanthrene	Tebuconazole
**CAS RN**	24365-47-7	83-79-4	534-52-1	85-01-8	107534-96-3
**MW ^(1)^ (g/mol)**	426.55	394.42	198.13	178.23	307.82
**LogK_OW_^(2)^**	−0.23	4.1	2.13	4.46	3.70 (exp.)
**Wsol ^(3)^ (mg/L)**	161.7 @ 25 °C	0.2 @ 20 °C, 0.17 @ 25 °C ^(4)^	198/678.4 @ 25 °C	0.677/1.15 @ 25 °C	36 @ 20 °C
**Usage ^(1)^**	Protease inhibitor	Insecticide (acaricide)	Insecticide (fungicide, herbicide, and antibacterial drug)	Environmental pollutant	Triazole fungicide (and antibacterial drug)
**Mode of action**	Inhibitor of cathepsin L	Inhibition of Complex I of the electron transport chain	Decoupler	Narcotic toxicity	Inhibition of sterol biosynthesis
**Parameters (lethal effects)**	a = 100 b = 3.2691 c = x50 = 353.76 R^2^ = 0.9691	a = 100 b = 2.2794 c = x50= 0.0376 R^2^ = 0.8049	a = 100 b = 56.5262 c = x50 = 3.3646 R^2^ = 0.8374	a = 100 b = 4.1481 c = x50 = 3.9560 R^2^ = 0.9146	a = 100 b = 12.9412 c = x50 = 62.4523 R^2^ = 0.9275
**Sublethal effects**	Not significantly different from lethal effect “curve”	Not significantly different from lethal effect “curve”	Not significantly different from lethal effect “curve”	Not significantly different from lethal effect “curve”	a = 100 b = 1.34 c = x50 = EC50 = 19.09 R² = 0.989

^(1)^ :http://www.chemspider.com (October 2018); ^(2)^ :http://www.epa.gov/opptintr/exposure/pubs/episuite.htm, Vers. 4.11; ^(3)^ : www.ACDlabs.com, Vers.12, 2008; ^(4)^ : Yalkowsky et al. (2010) [52]. Used data model fit formula for all six datasets: Hill, three parameters with y = a*x^b/(c^b+x^b); a = maximum, b = slope and c = x50 = LC50 and EC50 for tebuconazole (µmol/L).

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
