# Peer review of "Effects of Five Substances with Different Modes of Action on Cathepsin H, C and L Activities in Zebrafish Embryos"

_ijerph, 2019, doi:10.3390/ijerph16203956_

Round 1

Reviewer 1 Report

The paper by Kuster et al. studied effects of five chemicals with different MOAs on cathepsins in zebrafish embryos. Overall, the paper is interesting. There are a number of points for improvements.

line 42 and 43, the sentence "effects of.... " is not clear and difficult to understand. It is suggested that this sentence should be deleted. line 50 and line 120, the term "teratogenicity assay" DarT is old and rather confusing. As the OECD test guideline has been approved in 2013 and the authors have referred to it, it is suggested that the authors should use the OECD test guideline term and the test can be referred to as FET. In relation to the FET, the authors indicated in the key words "3R animal test refinement method". This keyword is not completely correct. Refinement is one of the 3R and the FET is also considered to be a replacement of fish acute toxicity test. For details, the authors are suggested to refer to the review paper published in Chemosphere 2017, 186, 677-685. line 77, one "to" should be deleted. line 119 to line 134, DarT should be changed into FET. The test concentrations are not indicated. It is not clear whether solvent was used or not. line 127 and line 129, the statement "10 embryos per concentration and replicate" is not clear. In relation to figure 1, it is important to indicate why exposure of 48 hours was chosen. Table 1, there are two types of a, b c. It is important to use different symbols. It is suggested that LC50 should be clearly stated. The unit of LC50 should be used for mg/L. line 409-415 this paragraph should be in the introduction section. It is also a bit overlapped with the description in the introduction section. It should be deleted or incorporated into the introduction section.

Reviewer 2 Report

Comments on the manuscript:

“Effects of Five Substances with Different Modes of 2 Action on Cathepsin H, C and L Activities in 3 Zebrafish Embryos”

In this study, the authors evaluated the role of cathepsins such as biomarkers for toxicology in the zebrafish Danio rerio. After explaining the physiological roles of cathepsins, more especially in the yolk lysosomal process, they evaluated the enzyme activity of several cathepsins in different stages of Danio rerio embryos, at the beginning of development. The authors concluded that study does not support the use of cathepsins as general biomarkers but they support their use as biomarkers for identifying cathepsin inhibition.

This study is interesting and certainly useful. Nevertheless, the manuscript is sometimes hard to read. In conclusion, the manuscript needs to be improved . I have several remarks and corrections.

Line 48: don’t write “Hamilton-Buchanan 1822)” in italics.

Line 129: “Petri” instead “petri”.

Line 135: 2.3. Danio rerio Embryo Collection for Enzymatic Measurements during Development: this part of material and methods is hard to be understood

                - Lines 136, 137: How many embryos were used for each stage?

                - Line 140: If I well understood, 30 embryos of each stage were in 2mL Eppendorf tube?

It would be clearer if the authors first described the steps used, then the number of embryos in each tube, and then what was done with these embryos.

                - line 153: A table or organizational chart showing the distribution of embryos with stages under different conditions would be useful, indicating the replicates.

Line 158: few embryos: how many exactly?

Line 439: I did not find any reference between 59 and 6

References:

I did not find ref 50;

Line 439: I did not find any reference between 59 and 63

Check the references in the text.

Figures

Figure2: Part B of Figure 2 is announced in the legend, but no image is indexed B

Figure 3: Parts A and B of the figure are given in the legend but not in the figure

Round 2

Reviewer 2 Report

Comment for the 2nd version of the manuscript ““Effects of Five Substances with Different Modes of 2 Action on Cathepsin H, C and L Activities in 3 Zebrafish Embryos””

The manuscript has been improved but imperfections remain and some minor corrections are needed.

Lines 159, 194: “Petri” with a capital letter instead “petri”. (Petri is the name of a microbiologist)

Figure 2: the parts A and B of Figure are announced in the legend, but no image is indexed A or B

Figure 3: the parts A and B of Figure are announced in the legend, but no image is indexed A or B

Figures are given twice.

Author Response

Dear editor,

I did change all three things in the manuscript according to the reviewer 3.

The figures seemed to have been "broken" the last time. I did do the changes and attached the pdf file last time but this might have been overlooked. As I am using LibreOffice instead of Microsoft Offices this might have been the reason.

Now I included new figures with the letters A and B and hope they don't get mixed up in the uploading process. In case it did not work either please refer to my last revision in which I attached a pdf with the corrected two figures.

Sincerely,

Eberhard Küster
